# Exploitation of Marginal Hilly Land in Tuscany through the Cultivation of *Lavandula angustifolia* Mill.: Characterization of Its Essential Oil and Antibacterial Activity

**DOI:** 10.3390/molecules27103216

**Published:** 2022-05-17

**Authors:** Basma Najar, Luisa Pistelli, Filippo Fratini

**Affiliations:** 1Dipartimento di Scienza Agraria, Alimentari e Agro-ambientale, Università di Pisa, Via del Borghetto, 80, 56124 Pisa, Italy; 2Dipartimento di Farmacia, Università di Pisa, Via Bonanno Pisano, 6, 56126 Pisa, Italy; luisa.pistelli@unipi.it; 3Centro Interdipartimentale di Ricerca Nutraceutica e Alimentazione per la Salute (Nutrafood), Università di Pisa, 56124 Pisa, Italy; filippo.fratini@unipi.it; 4Dipartimento di Scienze Veterinarie, Università di Pisa, Viale delle Piagge 2, 56124 Pisa, Italy

**Keywords:** lavender essential oil, age effect, antimicrobial activity, gram-positive, gram-negative

## Abstract

*Lavandula angustifolia* Mill., known as one of the best essential oil-bearing plants, is an aromatic plant that is well cultivated in many Mediterranean regions due to its adaptability to variations in climatic and edaphic conditions. Therefore, its essential oil (EO) composition and its antimicrobial activity change as a consequence of abiotic and biotic factors. The chemical composition of *L. angustifolia* EO collected during four consecutive years of growth was one of the aims of this work. The volatile profile evidenced the prevalence of linalool and linalool acetate even though they switched their positions according to age. Plants in their first year were characterized by a high amount of sesquiterpene compounds (22.1% of the identified fraction). This percentage decreased during plant growth, not representing more than 5.3% in the fourth year. It is interesting to note that both the third- and fourth-year plants showed a content of monoterpenes that exceeded 90% of the total identified constituents. The EO extracted from the oldest plants evidenced higher activity on the studied strains, with more sensitivity on the Gram-positive ones. Tuscan lavender EO, especially that obtained from the four-year-old plants, is of great interest for its potential industrial applications and constitutes an example for the valorization of marginal Tuscan land and good-quality production.

## 1. Introduction

The global essential oils market has increased exponentially in the last decade, and it is estimated to be valued USD 10.3 billion in 2021 (https://www.marketsandmarkets.com/Market-Reports/essential-oil-market-119674487.html, accessed on 24 January 2022), of which USD 38.3 million is attributable to lavender oil (https://www.globenewwire.com/news-release/2020/09/29/2100919/0/en/Global-Lavender-Oil-Industry.html, accessed on 24 January 2022). The latter is known as the best essential oil-producing plant primarily used in cosmetics, personal care products, and herbal drugs [1]. Three of these plants belong to *Lavandula* spp., cultivated for the commercial production of their essential oils (*Lavandula angustifolia* Mill., *Lavandula × intermedia* Emeric ex Loisel, and *Lavandula latifolia* Medicus) [2], even though *L. angustifolia* is the most important species of this genus. Its essential oil remains highly valued due to its attractive fragrance and low camphor content, even though its oil yields are less than the yields of spike oil (from *L. latifolia*) or lavandin oil (from *Lavandula × intermedia*) [3]. The natural distribution of the subspecies *angustifolia* is in southern France and northern Italy, while sp. *pyrenaica* is concentrated in the Pyrenees [4]. *L. angustifolia* is one of the most cultivated plant species in Italy, covering an area of about 137 ha of marginal and abandoned lands, due to its resistance to climatic changes and diseases. The value of its agronomic production amounts to EUR 5800/ha in reference to essential oil production and its market [5]. In the last five years, the Tuscan regions have improved the cultivation of lavender and lavandin following the results of different projects dedicated to the organic cultivation of aromatic plants and to exploitation of hilly areas (Flora–Flora aromatica S. Luce e la Valle dei Profumi; PIF–PSR Toscana, 2015–2020).

Many compounds have been identified and characterized in the *Lavandula* spp EOs, leading to the definition of international standards for the quality of this EO. The main international standard for lavender (*Lavandula angustifolia*) was set out by the International Standards Organization (ISO 3515:2002/Cor 1:2004, https://www.iso.org accessed on 24 January 2022 and NF ISO 3515:2004 (T75-301), https://www.boutique.afnor.org accessed on 24 January 2022) and in therapeutic pharmacopeias. Both organizations agreed that lavandulol and lavandulyl acetate are key lavender EO constituents, even though they are present in a minor percentage in comparison to linalool and linalool acetate, which are the main components. The cited standards confirmed that low levels of camphor and eucalyptol are important in lavender essential oil to differentiate it from lavandin oil. Previous works showed a broad spectrum of the biological activities of this EO, which is used for its sedative, anti-inflammatory, antioxidant, antimicrobial, and antifungal properties, in addition to its use as an insecticide and larvicides, and as an anticancer remedy and food additive [6,7,8].

Currently, many literature reports focus on the effect of abiotic stress on the composition of lavender essential oil [9,10,11], as well as genetic factors [12], while few works have dealt with the effect of the ontogenetic development [13,14]. However, other essential oil-bearing plants have demonstrated the influence of the endogenous “plant age” factor on EO yield and composition [15,16,17]. This work reports for the first time the trend of the composition of essential oils from organic *Lavandula angustifolia* cultivated in hilly marginal land in Tuscany (Molazzana, Garfagnana, Lucca, Italy), which was collected in four different years of its ontogenetic growth. The antibacterial properties of the lavender EOs were also assessed.

## 2. Results and Discussion

### 2.1. Phytochemical Analyses

During the years of the experiment, an increase in the EO yield was noted, even though a non-significant decrease was observed in the four-year-old plants (Table 1). Its amounts ranged from very low in the first year to reaching more than 1.3% in the last two years of the experiment. This disagreed with previous results performed on the Croatian and Egyptian species where the yields were 0.9% and 1%, respectively [18,19]. Overall, 55 compounds were identified in the essential oils of the studied species through the four years of growth, with the percentage of identification varying from 97.5% during the first year to 100% identification in the fourth year. Oxygenated monoterpene was the main class in all the samples, and its amount increased with age, except in the fourth year, where a slight decrease was observed (66.1% in the first year vs. 83.4% in the third year). In more detail, linalool and linalool acetate were the prevailing compounds even though their amount changed during the four years of growth. In fact, in the first-year linalool acetate, monoterpene ester dominated (26.3%) followed by linalool, a monoterpene alcohol (19.3%). This order changed in the following years, and linalool regained its dominant position, reaching 34.2% in the fourth year. On the contrary, linalool acetate was shifted into the second position in the next three years, and its amount varied between 21.2%, 26.0%, and 19.4%, respectively. It is worth nothing that, with the plant growth, a quarter of the linalool acetate was lost, while the linalool percentage was increased by about 20% and 44%, reaching almost 78% when passing from the first, second, third, and fourth years, respectively.

Statistically speaking, both linalool (first year vs. fourth year and second year vs. fourth year) and linalool acetate (first year vs. fourth year and third year vs. fourth year) showed a significant difference in their percentages, but only linalool evidenced a high positive correlation with age (correlation coefficient: 0.9), which means that its amount increased with age. Such correlation was observed in two other characteristic compounds of lavender essential oil: 3-octanone (correlation coefficient: 0.9) and α-terpineol (correlation coefficient: 0.7).

In the literature, there are numerous contributions pointing out the abundance of linalool and linalool acetate in *L. angustifolia* [3,14,19,23,24].

The presence of terpinene-4-ol, a mono terpineol like linalool, among other characteristic compounds of lavender oil, was relevant, occurring in high amounts in the second and fourth years of growth (15.3% and 9.3%, respectively). It was also detected in both the first and third years, but in a smaller almost (5.4% and 5.7%, respectively). Contrarily, these latter years were characterized by a high level of lavandulyl acetate (6.1% and 10.0%, respectively). It is worth noting the decrease in sesquiterpenes, both hydrocarbon (SH) and oxygenated (OS) ones, over the years and, consequently, the reduction in β-caryophyllene and tau-cadinol by of 50% and 91%, the main constituents of SH and OS, respectively.

The investigation on the growth-age effect in lavender oil composition was reported in a study performed by our team in 2017 [13] where the researchers evidenced that the amount of linalool decreased over the years while linalool acetate increased. This finding was confirmed by the work of Détàr [14] but disagreed with the results reported here where an upheaval of this behavior was observed. The activation of the transcription of linalool synthase in flowers [25], the growth age [9], or the climatic condition [3] may be responsible of these changes.

To standardize the EO composition of lavender, both the European Pharmacopeia (PH-Eur) and the ISO specify certain characteristics and establish the limits for the composition of these oils (Table 2). It is interesting to note that the composition respected the limits of both the PH-Eur and ISO only in the first year, except for linalool, the amount of which was slight less than the recommended value. It seems that aging worsens the quality of the product given that linalool acetate, terpinene-4-ol, and α-terpineol showed different levels than those predicted by the standard norms in the second and fourth years, while the third year evidenced a high amount of lavandulyl acetate and α-terpineol. Rocha and co-workers [17], investigating the age effect on *Cymbopogon citratus* (DC.) Stapf., pointed out that the age significantly affected the chemical composition of the EO. In lavandin CAS08, the linalool amount decreased in the eldest plants in comparison to the younger ones [26].

PCA1 (the direction explaining the maximum variance (51.8%)) showed a clear difference between the first year of growth, which is plotted at a negative value of PC1 (top-left quadrant) (Figure 1A). This position was due to the presence of sesquiterpene compounds, with variables with high negative loading on PC1. The third and fourth year were positioned at negative values of PC2 (bottom-right quadrant), with the axis explaining 27.7% of the total variance. These ages evidenced a high content of linalool, variable whose loading is positive along PC1 and negative on PC2 (Figure 1B). This is inverse to what was observed for the second year, which has positive values along PC2.

The results of the two-way hierarchical cluster analysis (Figure 2), performed on the volatile compounds that were present with a percentage >0.5%, agreed perfectly with the PCA results. In fact, the HCA clustered the age of growth (years) into two different groups: A and B. Group A was homogenous and included the samples of the first year. Group B was divided in two subgroups: Ba and Bb. Subgroup Ba included samples of the second year, while subgroup Bb comprised the third and fourth years of growth.

### 2.2. Antibacterial Activities

The antibacterial activities of the EOs were assessed on three Gram-positive bacteria strains: *Staphylococcus aureus* ATCC 6538, *Enterococcus faecalis* VAN B V 583 E, and *Listeria monocytogenes*, together with three Gram-negative bacteria strains: *Pseudomonas aeruginosa* ATCC 27853, *Escherichia coli* ATCC 15325, and *Salmonella enterica* ser. Typhimurium ATCC 14028. The results are represented in Table 4 and Table 5.

Considering the age of the plants from which the oil was extracted, the ones that showed a higher inhibitory and bactericidal activity were those of the fourth year, followed by (in decreasing order) those of the third year, the second year, and finally the first year.

The best inhibiting results, as widely found in the literature [27,28], were against Gram-positive bacteria, with MIC values as low as 1:16 against all three strains tested from the EO of first-year plants to values of 1:128 for *Listeria monocytogenes* for the EOs of fourth-year plants.

Gram-negative bacteria were found to be less inhibited by oils derived from plants in each year with maximum MIC values of 1:32 for *Escherichia coli* for fourth-year plants.

The most sensitive bacterium was *Listeria monocytogenes* (a maximum MIC of 1:128 and MBC of 1:64 for the EOs of fourth-year plants), while the most resistant was *Pseudomonas aeruginosa* (a maximum MIC of 1:16 and MBC of 1:8 for the EO of fourth-year plants).

Comparing these data with the GC–MS analyses, the trend of increased inhibitory efficacy against bacteria demonstrated as the plants aged is superimposed by the trend of an increasing relative percentage of linalool (19.3, 23.1, 27.6, and 34.2 for each year), which was highly correlated with the ATCC15325 activity (correlation coefficient: 0.9) (Table 6). Linalool, a pleasant floral scent compound that is widely used in cosmetics and the pharmaceutical and food industries [29] has been noted for its antibacterial activity against *Staphylococcus aureus* NCTC 10788, *Pseudomonas aeruginosa* NCTC 12924, and *Escherichia coli* NCTC 12923 [30]) and has shown a significant effect on *Pseudomonas fluorescens* with 1.25 and 2.5 μL/mL of the MIC and MBC, respectively [31].

The increase in the amount of α-terpineol was correlated positively with the EO activity (meaning that when its amount increased, the bacteria became more sensitive to this essential oil), even though ATCC6538 was shown to be more sensitive to the increase in the latter in the EO (correlation coefficient: 0.9, Table 6). This compound was reported to have a good effect on Gram-positive bacteria [32]. Furthermore, geraniol is a compound with fast action (time taken to produce significant action on bacteria) and can inactivate organisms such as *E. coli* and *Salmonella* in 5 min [33].

## 3. Materials and Methods

### 3.1. Plant Material and Cultivation

The seeds were bought from a local producer in Tuscany (La Semeria di Stefano Baldassini, Via Filippo Turati 271/A, 54011 Aulla, Italy). Since seeds require certain conditions to germinate, as a first step, we proceeded to store them for two months in a refrigerator in a container containing soil to simulate a sort of winter. Starting from the end of February (2016), sowing in the seedbed was performed. After some months, when the seedlings were big enough (8−10 cm), they were transferred first into single pots and then into bigger pots until planting in the spring of the following year. Cultivation was performed in an open field in the Azienda Agricola “La Rosa di Sassi” (Molazzana-Garfagnana, Lucca province, Tuscany, Italy, 44.096127, 10.394367) by planting the one-year-old seedlings on the ground in rows, maintaining 1.5 m between the rows and with a 60 cm distance between the seeds in the row. The land was characterized by clayey and weakly alkaline soil; it is well drained because it is sloping. No treatment or fertilization was carried out since it was previously pastureland; only plowing and milling had been performed. The flowers were harvested in the best balsamic period (always in the second and/or third week of July during the hot hours) and hand pruned. No drying pre-treatment was carried out; thus, the plant material was weighed to calculate the yield and immediately distilled. The essential oil yields are reported in Table 1.

### 3.2. Phytochemical Survey

#### 3.2.1. Essential Oil Extraction

Fresh aerial parts (50 g) collected during the four years of cultivation were hydrodistilled to obtain the EO using a Clevenger apparatus as recommended by the European Pharmacopeia. The experiment was performed in duplicate, and the time of the extraction was two hours. By the end of each distillation, the EOs were collected in glass vials and kept in a fridge at 4 °C until the analyses.

#### 3.2.2. GC–MS Analyses

The EOs were diluted to 5% and then analyzed by GC–MS. The equipment description and conditions of use, together with the identification method for the compounds, have been reported in previous studies [34].

### 3.3. Antimicrobial Activity

The EOs were tested for antimicrobial activity against the following bacterial strains: *Staphylococcus aureus* ATCC 6538, *Enterococcus faecalis* VAN B V583E, *Listeria monocytogenes* ATCC 7644, *Pseudomonas aeruginosa* ATCC 27853, *Escherichia coli* ATTC 15325, and *Salmonella enterica* serovar Typhimurium ATCC 14028. Bacterial strains stored at −80 °C in glycerol suspension were sowed on tryptic soy agar (TSA) (Oxoid, Milan, Italy) and incubated overnight at 37 °C. Subsequently, one colony from these cultures was inoculated in a brain heart infusion broth (BHI) (Oxoid, Milan, Italy) and incubated at 37 °C for 24 h with shaking in order to obtain freshly cultured microbial suspensions.

The EOs’ MIC (minimum inhibitory concentration) and MBC (minimum bactericidal concentration) values for each strain were determined using the two-fold serial microdilution method according to the protocol in [35], with some modifications previously reported [36]. Both the MIC and MBC results were expressed as v/v and reported as mode values.

All the EOs were stored in the fridge at 4 °C and were subjected to microbial analysis for quality control before their use in the tests. The dilutions of each oil carried out in peptone water were spread onto plate count agar (PCA) (Oxoid, Milan, Italy), and these were enumerated after incubation at 30 °C for 72 h.

### 3.4. Statistical Analyses

The experimental data were expressed as mean values± standard error of determinations made in triplicates. For the multifactorial comparison, a correlation matrix of 342 variants in the dataset was used for the principal component analysis (PCA) and two-way hierarchical cluster analysis (HCA). Briefly, this matrix was used for the measurement of eigenvalues and eigenvectors in the PCA analysis where the plot was performed selecting the two highest principal components (PCs). The distance between samples was calculated using Ward’s method and square Euclidean distances for a two-way measurement. Correlations between the ages in years and the compounds, as well as between the antibacterial activity and compounds, were expressed as correlation coefficients. One-way analysis of variance (the ANOVA test) was performed in order illustrate significant differences between the means, and HSD Tukey’s test (*p* < 0.05) was applied to compare the mean value of the volatile compounds at different ages. The statistically significant differences induced by the age in years were assessed using PERMANOVA (Permutational Multivariate Analysis of Variance) test with the Bray–Curtis dissimilarity index, which were based on a distribution-free analysis of variance.

## 4. Conclusions

The chemical composition of lavender essential oils was firstly analyzed over a period of four years of growth in marginal land in Tuscany. Linalool acetate and linalool were the principal components of the lavender oils, even though their relative positions changed from the first year to the following three years. Although the quality of the essential oil obtained from the older plants did not meet the parameters required by the European Pharmacopoeia, it was obtained in higher yields in comparison with the results from previous studies reported in the literature. Furthermore, these oils had non-negligible antibacterial activity. Therefore, lavender has proven to be a successful model for cultivation in marginal areas such as Tuscany, and its EO may be of greater interest due to its potential applications in various industries according to its composition.

## Figures and Tables

**Figure 1 molecules-27-03216-f001:**
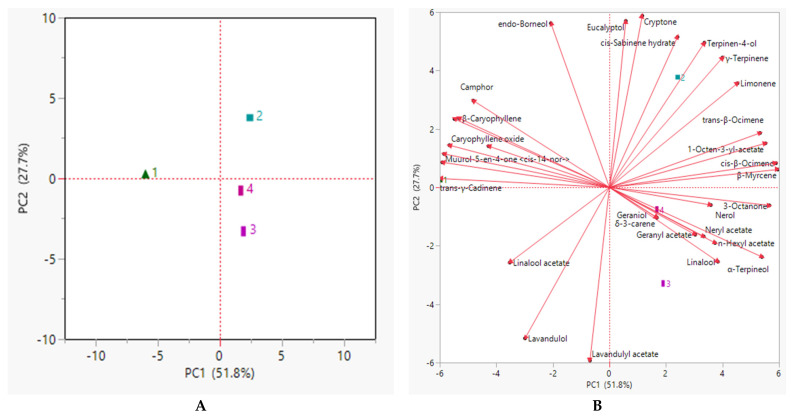
Principal component analyses of the essential oil composition of *Lavandula angustifolia* collected during four consecutive years. (**A**) Loading Plot; (**B**) Biplot.

**Figure 2 molecules-27-03216-f002:**
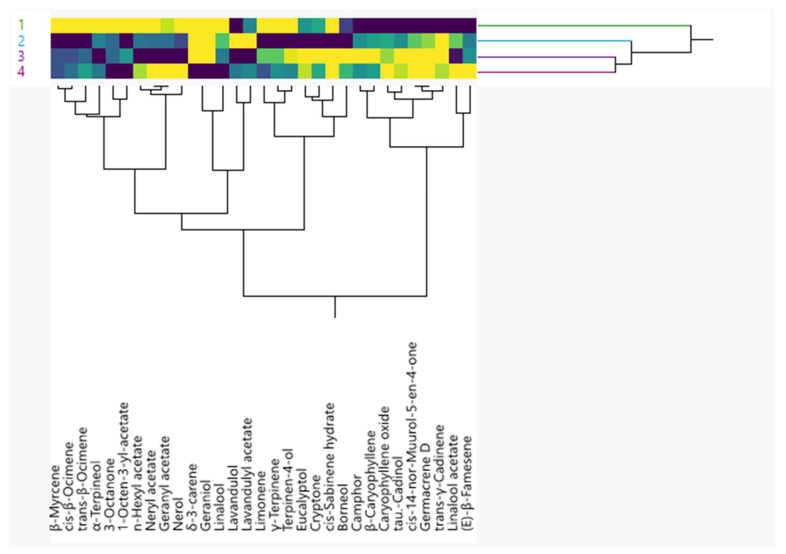
Two-way hierarchical cluster analysis performed using Ward’s methods for lavender EO. Even though the one-way PERMANOVA (Table 3) evidenced a significant difference between the years (*p* value = 0.0104 < 0.05 gradient), the pairwise test was unable to evidence the years between which these differences were noted.

**Table 1 molecules-27-03216-t001:** Essential oil composition of *Lavandula angustifolia* during the four years of growth and collection (mean ± SD).

Compounds	LRI^exp^	LRI^lit^	Class	1st Year	2nd Year	3rd Year	4th Year	*Correlation Coefficient* ^$^
				*Relative abundance (%)*				
α-thujene	929	930	mh	-^c^	0.2 ± 0.02^a^	0.1 ± 0.01^b^	0.2 ± 0.01^a^	
α-pinene	937	939	mh	0.1 ± 0.08^b^	0.4 ± 0.02^a^	0.1 ± 0.00^b^	0.3 ± 0.05^a^	
camphene	952	946	mh	0.1 ± 0.03^a^	0.2 ± 0.01^a^	-^b^	0.1 ± 0.02^a^	
1-octen-3-ol	979	978	nt	-	-	-	0.3 ± 0.00	0.8
3-octanone	986	983	nt	0.5 ± 0.09^b^	0.8 ± 0.04^ab^	0.8 ± 0.07^a^	0.9 ± 0.06^a^	0.9
β-myrcene	991	990	mh	0.4 ± 0.07^b^	1.0 ± 0.03^a^	0.9 ± 0.10^a^	0.9 ± 0.06a	0.7
3-octanol	995	991	nt	0.2 ± 0.03^a^	-^b^	-^b^	-^b^	−0.7
butyl butanoate	996	994	nt	-^c^	0.2 ± 0.01^a^	-^c^	0.2 ± 0.01^b^	
α-phellandrene	1005	1002	mh	-^b^	0.1 ± 0.00^a^	-^b^	-^b^	
*n*-hexyl acetate	1012	1009	nt	0.2 ± 0.02^b^	0.7 ± 0.04^ab^	1.1 ± 0.18^a^	0.3 ± 0.04^b^	
α-terpinene	1017	1017	mh	-^b^	0.1 ± 0.00^a^	-^b^	-^b^	
*o*-cymene	1022	1026	mh	0.2 ± 0.01^a^	0.2 ± 0.00^a^	-^b^	0.2 ± 0.05^a^	
limonene	1030	1029	mh	0.3 ± 0.03^b^	0.7 ± 0.06^a^	0.4 ± 0.05^b^	0.6 ± 0.07^a^	
δ-3-carene	1031	1030	mh	-^b^	-^b^	-^b^	0.5 ± 0.20^a^	0.8
eucalyptol	1032	1031	om	1.0 ± 0.03^b^	2.0 ± 0.01^a^	0.5 ± 0.06^c^	0.6 ± 0.02^c^	
*cis*-β-ocimene	1038	1037	mh	3.1 ± 0.04^b^	4.2 ± 0.17^a^	4.0 ± 0.52^a^	3.9 ± 0.25^a^	
*trans*-β-ocimene	1049	1050	mh	2.7 ± 0.02^c^	4.0 ± 0.18^a^	3.6 ± 0.08^ab^	3.3 ± 0.14^b^	
γ-terpinene	1060	1059	mh	0.1 ± 0.01^c^	0.6 ± 0.01^a^	0.2 ± 0.01^c^	0.3 ± 0.02^b^	
*cis*-sabinene hydrate	1066	1070	om	0.2 ± 0.00^b^	0.5 ± 0.02^a^	0.2 ± 0.01^b^	0.2 ± 0.04^b^	
*cis*-linalool oxide (furanoid)	1074	1072	om	-^c^	0.2 ± 0.01^a^	0.1 ± 0.01^b^	0.1 ± 0.02^b^	
terpinolene	1088	1088	mh	0.1 ± 0.01^b^	0.4 ± 0.01^b^	0.3 ± 0.04^b^	0.3 ± 0.03^b^	
linalool	1099	1096	om	19.3 ± 0.54^b^	23.1 ± 2.63^b^	27.7 ± 1.80^ab^	34.2 ± 1.70^a^	0.9
1-octen-3-yl-acetate	1111	1112	nt	0.6 ± 0.01^b^	1.1 ± 0.21^a^	0.9 ± 0.20^a^	1.1 ± 0.22^a^	
3-octanol acetate	1125	1123	nt	0.1 ± 0.00^a^	-^b^	0.1 ± 0.01^a^	0.1 ± 0.03^a^	
camphor	1145	1146	om	0.5 ± 0.01^a^	0.3 ± 0.01^c^	0.1 ± 0.01^d^	0.3 ± 0.00^b^	
endo-borneol	1167	1169	om	1.4 ± 0.15^a^	1.6 ± 0.26^a^	0.5 ± 0.10^b^	0.9 ± 0.22^ab^	−0.7
(3*E*,5*Z*)-1,3,5-undecatriene	1174	1173	nt	-^b^	-^b^	0.1 ± 0.01^a^	-^b^	
lavandulol	1177	1169	om	0.8 ± 0.06^a^	-^b^	0.8 ± 0.13^a^	0.6 ± 0.08^a^	
terpinen-4-ol	1177	1177	om	5.4 ± 0.20^c^	15.3 ± 0.77^a^	5.7 ± 0.73^c^	9.3 ± 0.57^b^	
*p*-cymen-8-ol	1183	1182	om	-^b^	0.1 ± 0.00^a^	-^b^	-^b^	
cryptone	1184	1185	om	0.3 ± 0.02^b^	0.5 ± 0.03^a^	0.2 ± 0.03^b^	0.3 ± 0.02^b^	
α-terpineol	1189	1188	om	1.4 ± 0.02^b^	4.0 ± 0.54^a^	5.6 ± 0.65^a^	4.2 ± 0.41^a^	0.7
*n*-hexyl butyrate	1192	1192	nt	0.2 ± 0.05^b^	0.4 ± 0.05^a^	-^c^	0.3 ± 0.04^a^	
nerol	1228	1229	om	0.4 ± 0.01^b^	0.7 ± 0.03^ab^	0.8 ± 0.16^a^	0.4 ± 0.08^ab^	
cumin aldehyde	1239	1241	om	0.2 ± 0.00^a^	0.2 ± 0.00^a^	0.1 ± 0.01^c^	-^b^	−0.8
geraniol	1253	1254	om	-^b^	-^b^	-^b^	1.0 ± 0.15^a^	0.7
linalool acetate	1257	1254	om	26.3 ± 0.62^a^	21.2 ± 1.22^ab^	26.0 ± 1.98^a^	19.4 ± 1.28^b^	
bornyl acetate	1285	1288	om	0.2 ± 0.01^ab^	0.3 ± 0.01^a^	0.1 ± 0.08^b^	0.1 ± 0.02^ab^	
lavandulyl acetate	1304	1290	om	6.1 ± 0.08^a^	0.9 ± 0.00^b^	10.0 ± 2.14^a^	6.1 ± 0.74^a^	
neryl acetate	1364	1361	om	0.9 ± 0.02^c^	1.4 ± 0.02^b^	1.8 ± 0.52^a^	0.9 ± 0.35^c^	
geranyl acetate	1382	1381	om	2.0 ± 0.05^c^	2.7 ± 0.11^b^	3.3 ± 0.82^a^	1.9 ± 0.23^c^	
*n*-hexyl hexanoate	1384	1381	nt	0.1 ± 0.00^a^	-^b^	-^b^	-^b^	
β-caryophyllene	1419	1419	sh	7.0 ± 0.08^a^	3.5 ± 0.18^b^	1.3 ± 0.21^c^	3.5 ± 0.09^b^	−0.7
*trans*-α-bergamotene	1436	1434	sh	0.2 ± 0.01^a^	-^b^	-^b^	-^b^	−0.8
α-humulene	1454	1455^§^	sh	0.2 ± 0.00^a^	0.1 ± 0.01^b^	-^c^	-^c^	−0.9
(*E*)-β-farnesene	1457	1456	sh	2.9 ± 0.07^a^	2.0 ± 0.08^b^	1.8 ± 0.03^b^	0.6 ± 0.01^c^	−1
germacrene D	1481	1485	sh	1.4 ± 0.05^a^	0.4 ± 0.00^b^	0.3 ± 0.00^b^	0.3 ± 0.01^b^	
γ-amorphene	1496	1495	sh	-^b^	0.3 ± 0.01^a^	-^b^	-^b^	−0.8
*trans*-γ-cadinene	1514	1513^§^	sh	1.5 ± 0.06^a^	-^b^	-^b^	0.1 ± 0.00^b^	−0.7
caryophyllene oxide	1581	1583	os	2.1 ± 0.13^a^	0.8 ± 0.19^b^	0.5 ± 0.06^b^	0.3 ± 0.10^b^	−0.9
1,10-di-*epi*-cubenol	1619	1619	os	0.4 ± 0.02^a^	-^b^	-^b^	-^b^	−0.8
1-*epi*-cubenol	1627	1628	os	-^b^	0.1 ± 0.00^a^	-^b^	-^b^	
tau-cadinol	1640	1640	os	5.7 ± 0.35^a^	2.1 ± 0.20^b^	0.1 ± 0.00c	0.5 ± 0.21c	−0.9
*cis-*14-nor-muurol-5-en-4-one	1689	1689	os	0.7 ± 0.06^a^	0.1 ± 0.02^b^	-^c^	-^c^	−0.9
10-peroxy-murolan-3,9(11)-diene	1730	1729*	os	0.1 ± 0.01^a^	-^b^	-^b^	-^b^	−0.8
hexahydrofarnesyl acetone	1845	1845	ac	0.1 ± 0.02^a^	-^b^	-^b^	-^b^	−0.8
**Yield (*w/v*)**				**VL^c^**	**1.16^b^**	**1.36^a^**	**1.32^a^**	
**Class of Compounds**				**1st Year**	**2nd Year**	**3rd Year**	**4th Year**	
monoterpene hydrocarbons (mh)				7.0 ± 0.08^d^	12.0 ± 0.85^a^	9.5 ± 0.74^c^	10.8 ± 0.65^b^	
oxygenated monoterpenes (om)				66.1 ± 1.22^d^	74.3 ± 0.85^c^	82.9 ± 1.01^a^	80.3 ± 2.10^b^	
sesquiterpene hydrocarbons (sh)				13.1 ± 0.27^a^	6.3 ± 0.28^b^	3.4 ± 0.24^d^	4.5 ± 0.27^c^	
oxygenated sesquiterpenes (os)				9.0 ± 0.57^a^	3.2 ± 0.41^b^	0.6 ± 0.06^c^	0.8 ± 0.25^c^	
apocarotenoides (ac)				0.1 ± 0.02^a^	-^b^	-^b^	-^b^	
non-terpene hydrocarbons (nt)				2.2 ± 0.11^c^	3.8 ± 0.37^a^	3.3 ± 0.10^b^	3.6 ± 0.20^a^	
**Total Identified**				**97.5 ± 0.30^b^**	**99.6 ± 0.02^a^**	**99.7 ± 0.01^a^**	**100.0 ± 0.00^a^**	

Data are reported as mean values (n = 3 ± SD); LRI exp: Linear Retention Index determined on HP-5MS capillary column; LRT lit: Linear Retention Index reported by Adams 2007 [20], * NIST 2014 [21], and ^§^ NIST Chemistry WebBook. [22]; VL: very low. The superscript uppercase letters (a–d) indicate statistically significant differences between the samples. The statistical significance of the relative abundances was determined by Tukey’s post hoc test, with *p* ≤ 0.05. ^$^: A statistical test used to explain a correlation between age and the amount of compounds.

**Table 2 molecules-27-03216-t002:** Comparative requirements for *L. angustifolia* essential oil according to the Ph.Eur. (10th Ed.) and ISO 3515:2002.

	Studied EOs				Analytical Requirements	
Component	1st Year	2nd Year	3rd Year	4th Year	PH-Eur	ISO 3515:2002 (Other Origin) ^a^
limonene	0.3	0.7	0.4	0.6	≤1%	≤1%
1.8-cineole^b^	1.0	2.0	0.5	0.6	≤2.5%	≤3%
β-phellandrene^b^		0.1			–	≤1%
cis-β-ocimene	3.1	4.2	4.0	3.9	–	1–10%
trans-β-ocimene	2.7	4.0	3.6	3.3	–	0.5–6%
3-octanone	0.5	0.8	0.8	0.9	0.1–5%	≤3%
camphor	0.5	0.3	0.1	0.3	≤1.2%	≤1.5%
linalool	**19.3**	23.1	27.6	34.2	20–45%	20–43%
linalool acetate	26.3	**21.2**	26.0	**19.4**	25–47%	25–47%
terpinene-4-ol	5.4	**15.3**	5.7	**9.3**	0.1–8%	≤8%
lavandulyl acetate	6.1	0.9	**10.0**	6.1	≤0.2%	≤8%
lavandulol	0.8		0.8	0.6	≤0.1%	≤3%
α-terpineol	1.4	**4.9**	**5.6**	**4.2**	≤2%	≤2%

^a^ The ISO provides different specifications depending on origin. ^b^ 1.8-cineole and β-phellandrene can be coeluted. **Bold font**: Divergence from the PH-Eur and/or ISO 3515:2002; Council of Europe. European Pharmacopoeia, 10th ed.; Council of Europe: Strasbourg, France, 2020.

**Table 3 molecules-27-03216-t003:** One-way PERMANOVA test performed on whole-EO composition (using Bray-Curtis as the similarity index).

Permutation N	9999
Total sum of squares	0.1687
Within-group sum of squares	0.00755
F	28.45
*p* (same)	0.0104

**Table 4 molecules-27-03216-t004:** Mode of minimum inhibitory concentration (MIC) of lavender essential oils (EOs) against the tested ATCC microorganisms.

	Microorganisms	1st Year EO	2nd Year EO	3rd Year EO	4th Year EO
*Gram-positive*	*Staphylococcus aureus* ATCC 6538	1:16	1:32	1:64	1:64
	*Enterococcus faecalis* VAN B V 583 E	1:16	1:16	1:32	1:32
	*Listeria monocytogenes* ATCC 7644	1:16	1:16	1:32	1:128
*Gram-negative*	*Pseudomonas aeruginosa* ATCC 27853	1:8	1:8	1:16	1:16
	*Escherichia coli* ATCC 15325	1:8	1:8	1:16	1:32
	*Salmonella enterica* ser. Typhimurium ATCC 14028	1:8	1:8	1:16	1:16

**Table 5 molecules-27-03216-t005:** Mode of minimum bactericidal concentration (MBC) of lavender essential oils (EOs) against the tested ATCC microorganisms.

	Microorganisms	1st Year EO	2nd Year EO	3rd Year EO	4th Year EO
*Gram-positive*	*Staphylococcus aureus* ATCC 6538	1:8	1:16	1:32	1:32
	*Enterococcus faecalis* VAN B V 583 E	1:8	1:8	1:16	1:16
	*Listeria monocytogenes* ATCC 7644	1:8	1:8	1:16	1:64
*Gram-negative*	*Pseudomonas aeruginosa* ATCC 27853	>1:8	>1:8	1:8	1:8
	*Escherichia coli* ATCC 15325	>1:8	>1:8	1:8	1:16
	*Salmonella enterica* ser. Typhimurium ATCC 14028	1:8	1:8	1:8	1:8

**Table 6 molecules-27-03216-t006:** Correlation coefficients of the MIC and the main compounds of lavender EO.

	ATCC 6538	VAN B V 583 E	ATCC 7644	ATCC 27853	ATCC 15325	ATCC 14028
(*E*)-β-famesene	−0.8	−0.9	−0.9	−0.8	−0.9	−0.8
3-octanone	0.9	0.6	0.6	0.7	0.8	0.7
camphor	−0.9	0.0	0.0	−0.6	−0.5	−0.6
caryophyllene oxide	−1.0	−0.5	−0.5	−0.7	−0.7	−0.7
*cis*-β-ocimene	0.9	0.3	0.3	0.4	0.4	0.4
geraniol	0.4	1.0	1.0	0.6	0.8	0.6
germacrene D	−1.0	−0.3	−0.3	−0.6	−0.6	−0.6
linalool	0.7	0.8	0.8	0.8	0.9	0.8
tau-cadinol	−1.0	−0.4	−0.4	−0.8	−0.8	−0.8
*trans*-γ-cadinene	−1.0	−0.3	−0.3	−0.5	−0.5	−0.5
α-terpineol	0.9	0.2	0.2	0.7	0.7	0.7
β-caryophyllene	−0.9	−0.1	−0.1	−0.7	−0.6	−0.7
β-myrcene	0.9	0.4	0.4	0.5	0.5	0.5
δ-3-carene	0.4	1.0	1.0	0.6	0.8	0.6

## Data Availability

The original dataset is available upon request from the authors.

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
