# Peer review of "Exploitation of Marginal Hilly Land in Tuscany through the Cultivation of *Lavandula angustifolia* Mill.: Characterization of Its Essential Oil and Antibacterial Activity"

_molecules, 2022, doi:10.3390/molecules27103216_

Round 1

Reviewer 1 Report

This ms studied its  essential oil  composition and its  antimicrobial activity change during  Lavandula angustifolia ontogenetic development. The results revealed that the contents and chemical composition of L. angustifolia EO the regulation of changes during four consecutive years of growth, further antibacterial experiment data showed that the EO extracted from the oldest plants evidenced higher activity on the studied strains with more sensitivity on the gram-positive ones. The results are interesting and provide theoretical and practical guidance for quality evaluation of Lavandula angustifolia for its potential  industrial  application. However, there are still some questions need to be cleared and expained,some data need be added.

  1. The effect of development stages on essential oil content and components should be introduced in introduction section(please citing publication: The variation in essential oils composition,  phenolic  acids  and  flavonoids  is  correlated  with  changes  in  antioxidant  activity  during Cinnamomum  loureirii bark growth, Arabian  Journal  of  2021,14:103249 https://doi.org/10.1016/j.arabjc.2021.103249)
  2. Line 48, 5800/ha, please adding unit after 5800
  3. Line 119-120, ms mentioned that, but disagreed the herein results where an upheaval, of this behaviour was observed. Please explained possible causes.
  4. Line 221-222, in materials and methods section, authors mentioned that thus the plant material was weigh to calculate the yield and immediately distilled. But I cannot find relative result and data. Please adding experimental data in results and discussion section.
  5. Line 181-182, this sentence: Gram-negative bacteria were found to be less inhibited by oils derived from plants in each year with maximum MIC values of 1:32 for Escherichia coli for fourth-year plants. Please check and confirm that maximum MIC values of 1:32 is right.

Author Response

Reviewer 1

Open Review

English language and style

(x) Extensive editing of English language and style required
( ) Moderate English changes required
( ) English language and style are fine/minor spell check required
( ) I don't feel qualified to judge about the English language and style

Yes

Can be improved

Must be improved

Not applicable

Does the introduction provide sufficient background and include all relevant references?

( )

( )

(x)

( )

Are all the cited references relevant to the research?

( )

(x)

( )

( )

Is the research design appropriate?

(x)

( )

( )

( )

Are the methods adequately described?

( )

(x)

( )

( )

Are the results clearly presented?

( )

(x)

( )

( )

Are the conclusions supported by the results?

(x)

( )

( )

( )

Comments and Suggestions for Authors

This ms studied its essential oil composition and its antimicrobial activity change during Lavandula angustifolia ontogenetic development. The results revealed that the contents and chemical composition of L. angustifolia EO the regulation of changes during four consecutive years of growth, further antibacterial experiment data showed that the EO extracted from the oldest plants evidenced higher activity on the studied strains with more sensitivity on the gram-positive ones. The results are interesting and provide theoretical and practical guidance for quality evaluation of Lavandula angustifolia for its potential industrial application. However, there are still some questions need to be cleared and expained, some data need be added.

  1. The effect of development stages on essential oil content and components should be introduced in introduction section (please citing publication: The variation in essential oils composition, phenolic acids and flavonoids is correlated with changes in antioxidant activity during Cinnamomum loureirii bark growth, Arabian Journal of 2021,14:103249 https://doi.org/10.1016/j.arabjc.2021.103249).

Answer: Thank you for your suggestion. We added a sentences in the introduction with relative references.  

  1. Line 48, 5800/ha, please adding unit after 5800

Answer: Thank you for your observation. The unit was added

  1. Line 119-120, ms mentioned that, but disagreed the herein results where an upheaval, of this behaviour was observed. Please explained possible causes.

Answer: Thank you for your comment. We tried to explain the change in behavior during the years citing 3 new works. Please see L122-L123.

  1. Line 221-222, in materials and methods section, authors mentioned that thus the plant material was weigh to calculate the yield and immediately distilled. But I cannot find relative result and data. Please adding experimental data in results and discussion section.

Answer: Results related to the EO yield are reported in table 1. We delated the sentence in L 248-L249 to avoid misunderstandings 

  1. Line 181-182, this sentence: Gram-negative bacteria were found to be less inhibited by oils derived from plants in each year with maximum MIC values of 1:32 for Escherichia coli for fourth-year plants. Please check and confirm that maximum MIC values of 1:32 is right.

Answer: The MIC value 1:32 represent the highest MIC obtained in gram-negative strains and it was found in E. coli.

Reviewer 2 Report

This review discussed the subject ‘Exploitation of marginal hilly land in Tuscany through the cultivation of Lavandula angustifolia Mill.: Characterization of its essential oil and antibacterial activity’ looks very interesting, yet it needs to be improved in a scientific manner about the subject matter. Grammatical and typographical errors especially punctuations should be carefully rechecked.

Abstract:

Line 15: as the best replace with one of

Line16: variation of climatic replaced with variation in climatic

Line 18: as well as to ontogenetic- remove its

Line 21: to age- remove the

Line 24: both the 3rd and 4th years showed a content of monoterpenes that exceed..

Line 28: constitutes

Keyword: remove gram-positive and gram-negative replace them with essential oil (EO) and terpenoids

  1. Introduction:

Line 38: Lavandula genus or spp

Line 41: species

Line 48-49: punctuation- In the last five years,

Line 53: Lavandula spp EOs

Line 58: lavender constituents/metabolites

Line 61: why low levels of camphor and eucalyptol are important in lavender EO? Explain.

Line 62: biological activities of this EO such as:

Line 68-69: This work reported for the first time on the trend of the chemical composition of Lavandula angustifolia essential oils from organic cultivated …. which were collected…

Line 70: efficacy to properties

  1. Result and Discussion:

Give a general Figure on EO Biosynthetic pathways in relation to detected chemical components through GCMS analysis. The Biosynthetic pathway must be relevant to the plant and chemical components found through the analysis conducted as well as should be explained scientifically in a few sentences.

Explain in detail why linalool and linalool acetate was chosen for the comparison in the variability with the relation to maturity age. Try to relate with biosynthesis.

Line 84: prevailed compounds even their order changed? variation in content? detail out.

Lime 105: compounds

Line 116-120: Detail out why. Give proper chemical explanation.

Line 125: Why does aging worsen the quality of product and how does it significantly affects the chemical compositions in EO? Detail out.

Line 135-142: High content of linalool leads to what? What is the possible explanation for this statement? Try to correlate the findings from PCA analysis with variation in chemical compositions, cultivation year, and climate. Give detailed chemical explanations for the outcome.

2.2 Antibacterial activities:

Positive control for comparisons? Please include in the respective Tables 4 and 5.

  1. Materials and methods:

3.1. Plant material and cultivation: Botanist who did the plant sample identifications? Plant Voucher specimen no?

3.2 Phytochemical survey to Phytochemical analysis

GC-MS analyses: Detailed the procedure even though it's reported previously in other publications. It is an important part of this research article.

  1. Conclusion:

Highlight the future direction of this research clearly and how this outcome could be beneficial.

References:

There are some references cited not accordingly to the journal format. Please check carefully.

Author Response

Reviewer 2

Open Review

English language and style

( ) Extensive editing of English language and style required
(x) Moderate English changes required
( ) English language and style are fine/minor spell check required
( ) I don't feel qualified to judge about the English language and style

Yes

Can be improved

Must be improved

Not applicable

Does the introduction provide sufficient background and include all relevant references?

( )

(x)

( )

( )

Are all the cited references relevant to the research?

(x)

( )

( )

( )

Is the research design appropriate?

(x)

( )

( )

( )

Are the methods adequately described?

( )

(x)

( )

( )

Are the results clearly presented?

( )

( )

(x)

( )

Are the conclusions supported by the results?

( )

( )

(x)

( )

Comments and Suggestions for Authors

This review discussed the subject ‘Exploitation of marginal hilly land in Tuscany through the cultivation of Lavandula angustifolia Mill.: Characterization of its essential oil and antibacterial activity’ looks very interesting, yet it needs to be improved in a scientific manner about the subject matter. Grammatical and typographical errors especially punctuations should be carefully rechecked. 

Abstract:

Line 15: as the best replace with one of

Answer: Done

Line16: variation of climatic replaced with variation in climatic

Answer: Done

Line 18: as well as to ontogenetic- remove its

Answer:Done

Line 21: to age- remove the

Answer: Done

Line 24: both the 3rd and 4th years showed a content of monoterpenes that exceed.

Answer: Done

Line 28: constitutes

Answer: Done

Keyword: remove gram-positive and gram-negative replace them with essential oil (EO) and terpenoids

Answer: we disagree with the referee suggestion to remove gram-positive and gram-negative words and replace them with EO and terpenoids, but we add ‘essential’ in the first key word (Lavender essential oil). Terpenoids are the main compounds in the EO thus is irrelevant to insert this word in keywords.

  1. Introduction:

Line 38: Lavandula genus or spp

Answer: Done

Line 41: species

Answer:Done

Line 48-49: punctuation- In the last five years,

Answer: Done

Line 53: Lavandula spp EOs

Answer: Done

Line 58: lavender constituents/metabolites

Answer: Done

Line 61: why low levels of camphor and eucalyptol are important in lavender EO? Explain.

Answer: A high amount of camphor is referred to lavandin EO (Lavandula x hybrida) and it is not requested for Lavandula angustifolia high quality EO

Line 62: biological activities of this EO such as:

Answer: The word was corrected

Line 68-69: This work reported for the first time on the trend of the chemical composition of Lavandula angustifolia essential oils from organic cultivated …. which were collected…

Answer: The sentence was corrected as suggested by the reviewer

Line 70: efficacy to properties

Answer: Done

  1. Result and Discussion:

Give a general Figure on EO Biosynthetic pathways in relation to detected chemical components through GCMS analysis. The Biosynthetic pathway must be relevant to the plant and chemical components found through the analysis conducted as well as should be explained scientifically in a few sentences.

Explain in detail why linalool and linalool acetate was chosen for the comparison in the variability with the relation to maturity age. Try to relate with biosynthesis.

Answer: Since many figures on the biosynthetic pathway of the main EO constituents are available in the search engines it seemed that it is not relevant to insert such figure.

Linalool and linalyl acetate are marker compounds for the quality control of lavender EO as required by the standardization bodies. That is why these constituents were chosen for the comparison in variability with the relationship with the maturity age

Line 84: prevailed compounds even their order changed? variation in content? detail out.

Answer: The change in order means change in amount (relative percentage): linalool acetate was the predominant compound in the 1st year to be replaced by linalool in the successive years. This was explained in the successive sentence.

Lime 105: compounds

Answer: Done

Line 116-120: Detail out why. Give proper chemical explanation.

Answer: Thank you for your comment. We tried to explain the change in behavior during the years of growth adding 3 new references. Please see L135-L138.

Line 125: Why does aging worsen the quality of product and how does it significantly affects the chemical compositions in EO? Detail out.

Answer: A sentence was added to better explain the sentence

Line 135-142: High content of linalool leads to what? What is the possible explanation for this statement? Try to correlate the findings from PCA analysis with variation in chemical compositions, cultivation year, and climate. Give detailed chemical explanations for the outcome. 

Answer: In PCA analysis we plotted only the chemical composition of lavender EO extracted during the four years of the experiment. Unfortunately, we didn’t have the climate data set. Linalool showed higher percentage in 3rd and 4th year, and this was observed in the scatter plot. The reason why linalool is present is high amount in these years may be due to the activation of linalool synthase as reported in reference 26. 

2.2 Antibacterial activities: 

Positive control for comparisons? Please include in the respective Tables 4 and 5.

Answer: Results of the positive control is omitted because they are not comparable in dosage (synthetic compounds are obviously the most effective than the natural compounds or phytocomplex as EO).

  1. Materials and methods:

3.1. Plant material and cultivation: Botanist who did the plant sample identifications? Plant Voucher specimen no?

Answer: The certified seeds were used to start Lavandula angustifolia cultivation thus no botanist identification was necessary to confirm the exact plant material.  The voucher is needed when the plant is spontaneous collected not when it is cultivated.

3.2 Phytochemical survey to Phytochemical analysis

GC-MS analyses: Detailed the procedure even though it's reported previously in other publications. It is an important part of this research article.

Answer: We did not report detail of the GC-MS analysis to avoid any plagiarism. Please see our previous papers (reference 38)

  1. Conclusion:

Highlight the future direction of this research clearly and how this outcome could be beneficial.

Answer: we pointed out that Lavandula angustifolia can be cultivated in marginal area without any agronomic request and we determined the best time (years old) to collect plants and distilled its EO to be used for different industries, according to their composition.

References:

There are some references cited not accordingly to the journal format. Please check carefully.

Answer: we carefully checked the reference and correct them.

Reviewer 3 Report

The manuscript outlines the trends of changes in the composition of EOs recovered from Lavandula angustifolia from the perspective of the influence of time (four different years) of the plant's  ontogenetic growth. The antibacterial activity of the EOs were also assessed.

The results obtained are interesting from the view point of the perspectives for lavender to be grown in marginal areas, and the wide spectrum of applications of the EO. 

The manuscript is well structured, the methods used are described appropriately, interesting results are presented.

Though no breakthrough new information is advocated still, the paper  merits publication.

Some minor revision and editing that concern mainly the English should be, however introduced before publishing, for example:

Lines 74-75: The sentence

The yield in essential oil increased through the four years even though an unsignificant decrease was observed in the plants of four years old

should be revised.

Line 96 - should be which means

Line 106 "Many works were present..." - should be revised to read, e.g  In the literature there are numerous contributions which...

Line 119: "... but disagreed ...." should be edited (disagree with...)

etc.

Author Response

Reviewer 3

Open Review

English language and style

( ) Extensive editing of English language and style required
(x) Moderate English changes required
( ) English language and style are fine/minor spell check required
( ) I don't feel qualified to judge about the English language and style

Yes

Can be improved

Must be improved

Not applicable

Does the introduction provide sufficient background and include all relevant references?

(x)

( )

( )

( )

Are all the cited references relevant to the research?

(x)

( )

( )

( )

Is the research design appropriate?

(x)

( )

( )

( )

Are the methods adequately described?

(x)

( )

( )

( )

Are the results clearly presented?

(x)

( )

( )

( )

Are the conclusions supported by the results?

(x)

( )

( )

( )

Comments and Suggestions for Authors

The manuscript outlines the trends of changes in the composition of EOs recovered from Lavandula angustifolia from the perspective of the influence of time (four different years) of the plant’s ontogenetic growth. The antibacterial activity of the EOs were also assessed.

The results obtained are interesting from the view point of the perspectives for lavender to be grown in marginal areas, and the wide spectrum of applications of the EO. 

The manuscript is well structured, the methods used are described appropriately, interesting results are presented.

Though no breakthrough new information is advocated still, the paper merits publication.

Some minor revision and editing that concern mainly the English should be, however introduced before publishing, for example:

Lines 74-75: The sentence

The yield in essential oil increased through the four years even though an unsignificant decrease was observed in the plants of four years old

should be revised.

Answer: The sentence was revised

Line 96 - should be which means

Answer: Done

Line 106 "Many works were present..." - should be revised to read, e.g  In the literature there are numerous contributions which...

Answer: Done

Line 119: "... but disagreed ...." should be edited (disagree with...)

Answer: Done

etc.

Round 2

Reviewer 2 Report

Thank you for carrying out the corrections suggested.

Keywords:

Even though the authors disagree with removing the proposed keywords which are gram-positive and gram-negative. I don’t see any relevance in maintaining the keywords since the term antibacterial is sufficient. Moreover, in my opinion, there must be a chemical term need to be added such as terpenoids or metabolites to represent the chemistry work done by the authors.

2.2 Antibacterial activities

May I know the basis for choosing the bacterial strains for the antibacterial study conducted? Any relation to the studied EO? Specific reason.

Finally, I would like to congratulate the authors on their work.